# Interest of Homodialkyl Neamine Derivatives against Resistant *P. aeruginosa*, *E. coli*, and β-Lactamases-Producing Bacteria—Effect of Alkyl Chain Length on the Interaction with LPS

**DOI:** 10.3390/ijms22168707

**Published:** 2021-08-13

**Authors:** Jitendriya Swain, Clément Dezanet, Hussein Chalhoub, Marie Auquière, Julie Kempf, Jean-Luc Décout, Marie-Paule Mingeot-Leclercq

**Affiliations:** 1Université Catholique de Louvain, Louvain Drug Research Institute, Pharmacologie Cellulaire et Moléculaire, Avenue E. Mounier 73, UCL B1.73.05, 1200 Brussels, Belgium; jitendriyas4@gmail.com (J.S.); hsein85@hotmail.com (H.C.); marie.auquiere@uclouvain.be (M.A.); 2Université Grenoble Alpes, CNRS, Département de Pharmacochimie Moléculaire, Rue de la Chimie, F-38041 Grenoble, France; clement.dezanet@gmail.com (C.D.); julie.kempf@hotmail.fr (J.K.); jean-luc.decout@univ-grenoble-alpes.fr (J.-L.D.)

**Keywords:** amphiphilic aminoglycosides, antibiotics, *P. aeruginosa*, ESBL, lipopolysaccharides, bacterial lipid membrane

## Abstract

Development of novel therapeutics to treat antibiotic-resistant infections, especially those caused by ESKAPE pathogens, is urgent. One of the most critical pathogens is *P. aeruginosa*, which is able to develop a large number of factors associated with antibiotic resistance, including high level of impermeability. Gram-negative bacteria are protected from the environment by an asymmetric Outer Membrane primarily composed of lipopolysaccharides (LPS) at the outer leaflet and phospholipids in the inner leaflet. Based on a large hemi-synthesis program focusing on amphiphilic aminoglycoside derivatives, we extend the antimicrobial activity of 3′,6-dinonyl neamine and its branched isomer, 3′,6-di(dimethyloctyl) neamine on clinical *P. aeruginosa*, ESBL, and carbapenemase strains. We also investigated the capacity of 3′,6-homodialkyl neamine derivatives carrying different alkyl chains (C7–C11) to interact with LPS and alter membrane permeability. 3′,6-Dinonyl neamine and its branched isomer, 3′,6-di(dimethyloctyl) neamine showed low MICs on clinical *P. aeruginosa*, ESBL, and carbapenemase strains with no MIC increase for long-duration incubation. In contrast from what was observed for membrane permeability, length of alkyl chains was critical for the capacity of 3′,6-homodialkyl neamine derivatives to bind to LPS. We demonstrated the high antibacterial potential of the amphiphilic neamine derivatives in the fight against ESKAPE pathogens and pointed out some particular characteristics making the 3′,6-dinonyl- and 3′,6-di(dimethyloctyl)-neamine derivatives the best candidates for further development.

## 1. Introduction

Antibiotic resistance is a worldwide concern, and a group of six pathogens with growing multidrug resistance and virulence, including *Enterococcus faecium*, *Staphylococcus aureus*, *Klebsiella pneumoniae*, *Acinetobacter baumannii*, *Pseudomonas aeruginosa*, and *Enterobacter* spp. named ESKAPE, is more and more worrying [1,2]. ESKAPE pathogens rank among the most prevalent causative agents in common health care-associated infections, such as pneumonia, surgical site infection, urinary tract infection, and bloodstream infection. They are associated with a very high risk of mortality, thereby resulting in increased health care costs [1,2,3].

The mechanisms of multidrug resistance exhibited by ESKAPE are broadly grouped into three categories, namely drug inactivation, modification of the target site where the antibiotic may bind, and reduced accumulation of drug either due to reduced permeability or by increased efflux of the drug [4]. Various innate and acquired resistance mechanisms are involved in drug resistance of *Pseudomonas aeruginosa* [5], one of the most concerning pathogens [4,5]. The innate mechanisms include the overexpression efflux pump as the MexAB-OprM pump system that are capable of removing antibiotics from the periplasmic space to the external environment, and low permeability of outer membrane mediated by the decreased expression and subsequent loss of outer membrane proteins like porine OprD [6,7]. Acquired resistance involves the acquisition of resistance gene or mutation in genes encoding porins, efflux pumps, penicillin-binding proteins, and chromosomal β-lactamase, all contributing to resistance to β-lactams, carbapenems, aminoglycosides, and fluoroquinolones [5]. Carbapenem resistant *A. baumannii* and *P. aeruginosa* [8] along with extended spectrum β-lactamase (ESBL) or carbapenem resistant *K. pneumoniae* and *Enterobacter* spp. are listed in the critical priority list of pathogens from the World Health Organization (WHO).

Conventional aminoglycosides, including neomycin B (NEO), gentamicin (GEN), tobramycin (TOB), and amikacin (AMK), are widely used to treat infections caused by *P. aeruginosa*. They corrupt the accuracy of initial codon selection and proofreading in genetic code translation and inhibit EF-G dependent translocation of mRNA and tRNAs leading to the synthesis of aberrant proteins [9,10,11]. These last may be inserted into the cell membrane, leading to altered permeability and further stimulation of aminoglycoside transport. To reach their target, aminoglycosides have to cross bacterial membranes. Passage of highly polar molecules as aminoglycosides across the outer membrane of Gram-negative bacteria is a self-promoted uptake process involving the drug-induced disruption of Mg^2+^ bridges between adjacent lipopolysaccharide (LPS) molecules. Penetration through porin channels is unlikely because of their large size (>600 Da) [12,13]. Subsequent transport of aminoglycosides across the cytoplasmic (inner) membrane is dependent upon electron transport and is termed energy-dependent phase I (EDP-I) [14]. 

Widespread clinical use of aminoglycosides strongly reduced their clinical efficacy through the selection of resistant bacteria. The current situation prompted the study and development of alternative agents with new target-like membranes and/or multiple targets. Several approaches including the classical antibiotic hybrid drugs or use of prodrugs have been developed [15,16]. In the former, compounds are expected to elicit their antibacterial action by inhibiting two drug targets. However, utilizing only a singular molecular entity at the same time is a difficult feat to achieve. In the latter, no linker/tether aimed to be cleaved had been designed. Our group developed a large hemi-synthesis program focusing on amphiphilic aminoglycoside derivatives [17] with the aim to overcome the bottleneck in the development of new antibiotics against *P. aeruginosa* characterized by a very poor permeability [18]. By adding hydrophobic side chains to the core of conventional aminoglycoside like neamine, it was suspected to increase intrabacterial accumulation and efficiency. Amphiphilic neamine derivatives showed a broad spectrum of activities against both Gram-negative and Gram-positive bacteria, including most of the antibiotic-resistant strains and minimal induction of drug resistance. Among all the synthesized neamine derivatives reported until now [19,20,21,22], the 3′,6-dinonyl neamine seemed to be the most promising in regard to broad-spectrum antibacterial activity and moderate cytotoxicity. By using membrane model systems and bacteria (*P. aeruginosa*), we demonstrated the interaction of 3′,6-dinonyl neamine with LPS located at the outer membrane [23] and with cardiolipin, found at both outer and inner membranes, leading to membrane permeabilization and depolarization [24]. By targeting cardiolipin bacterial microdomains mainly located at the cell poles, 3′,6-dinonyl neamine leads to their disassembly into cardiolipin clusters and relocation of cardiolipin domains, resulting in bacterial morphological changes likely due to impairment of the dynamics of cell-shape determining proteins like MreB [24].

In a previous work [21], aimed to fine-tune the lipophilic window for improving activity and decreasing cytotoxicity, we synthesized and characterized new 3′,6-dialkyl neamine derivatives with different alkyl chain lengths (C7–C11). They have an identical backbone and differ only in the hydrophobic groups attached at the 3′- and 6′-positions of the neamine core. On the other hand, the influence of steric hindrance (3′,6-di(dimethyloctyl) neamine versus 3′,6-dinonyl neamine) has been characterized through introduction of branching in the alkyl chains preserving a similar lipophilicity in comparison to the 3′,6-dinonyl derivatives [21].

The objective of the current work was threefold. We first explored the potential interest of the lead compound, 3′,6-dinonyl neamine and its branched isomer, 3′,6-di(dimethyloctyl) neamine, on clinical *P. aeruginosa* and ESBL strains as well as their ability to induce MIC increase for long-duration incubation. Second, to confirm the membrane as the primary target for homodialkyl neamine derivatives, we explored their capacity to accumulate in the bacteria and inhibit protein synthesis. Third, based on a rational structure–activity relationship, we aimed to finely characterize the specificity of the 3′,6-dinonyl- and 3′,6-di(dimethyloctyl)-neamine derivatives in their interaction with LPS from Gram-negative bacteria *P. aeruginosa* and the ability to permeabilize bacterial inner membrane through comparison to other 3′,6-homodialkyl neamine derivatives carrying different alkyl chains (3′,6-diheptyl (diHp), -dioctyl (diOc), -didecyl (diDe) and -diundecyl (diUd)).

Here, we demonstrate the high antibacterial potential of the amphiphilic neamine derivatives in the fight against ESKAPE pathogens and point out some particular characteristics making the 3′,6-dinonyl- and 3′,6-di(dimethyloctyl)-neamine derivatives the best candidate for further development.

## 2. Experimental Procedures

The homodialkyl neamine derivatives with different chain lengths from 7 to 11 and 3′-heptyl-6-(1pyrenyl)butyl neamine were synthesized by Decout and colleagues [21]. Propidium iodide (PI) was ordered from Invitrogen (Paisley, Scotland, UK). Cell-free LPS from *P. aeruginosa* was obtained from Sigma-Aldrich (St. Louis, MO, USA). BODIPY-TR-Cadaverine [BC] was obtained from Molecular Probes (Invitrogen, Carlsbad, CA, USA). All other analytical grade reagents were purchased from E. Merck AG.

### 2.1. Bacterial Strain and Growth Conditions

Trypticase soy agar (TSA) plate was used to grow *P. aeruginosa* ATCC 27853 at 37 °C. One colony of bacteria was suspended in Cation Adjusted–Müller Hinton Broth (Ca-MHB, Sigma-Aldrich) and incubated overnight at 37 °C on a rotary shaker (130 rpm). 

### 2.2. MIC (Minimal Inhibitory Concentration) Determination 

All strains were grown overnight at 37 °C on TSA petri dishes (BD Diagnostics, BD, Franklin Lakes, NJ, USA). MICs were determined by microdilution method (96 well plate) using a fresh culture of different bacteria in Ca-MHB, according to the recommendations of the Clinical and Laboratory Standards Institute (CLSI) 2020 for *P. aeruginosa* and *Enterobacteriaceae*. 

Resistance selection method was performed by the serial passage method as previously described [21].

### 2.3. Accumulation of Fluorescent Amphiphilic Neamine Derivative in P. aeruginosa

*P. aeruginosa* ATCC strain 27853 was cultivated in Ca-MHB. The day before the experiment, ATCC 27853 strains were suspended in the Ca-MHB and incubated at 37° and 130 rpm overnight. After adjusting the bacterial concentration (1.10^9^ bacteria/mL), 5 mL of the bacterial suspension was taken and incubated with 3′-heptyl-6-(1-pyrenyl)butyl neamine [21] at 1.7 μM, corresponding to half of the MIC on *P. aeruginosa.* The samples were incubated for 15, 30, 60 or 120 min at 37 °C with gentle stirring (130 rpm) or at 4 °C. Then, the samples were collected and centrifuged for 8 min at 2880× *g* at 4 °C. The supernatant was kept and the pellet was washed twice with 5 mL of cold PBS (4 °C). The pellet was suspended in 600 μL of cold water (4 °C), and 590 μL of bacterial suspension was sonicated (Qsonica sonicator, Newtown, CT, USA) for 10 s at an amplitude 15. In 96-well dishes, 50 μL of samples were deposited per well. The fluorescence intensity of each sample was measured at an excitation wavelength of 340 nm and an emission wavelength of 390 nm.

Results were standardized by a CFU (colony-forming unit). In total, 10 μL of the bacterial suspension was taken and added to 90 μL of water to get a 10-fold dilution. Then, 10 μL of the solution diluted 10 times was taken and added to 90 μL of water to obtain a 100-fold dilution. The operation was repeated until the desired dilution factor was obtained (Petri dishes containing less than 300 colonies), and 50 μL of this dilute solution was placed on petri dishes prepared with TSA agar (trypto casein soy). They were put in the incubator overnight. The next day, the colonies were counted and results expressed in CFU/mL. The accumulation factor represents the bacterial concentration divided by the extra bacterial concentration.

### 2.4. Evaluation of Protein Synthesis—Luciferase Assay

To assess the capacity of 3′6-dinonyl neamine and 3′-heptyl-6-(1-pyrenyl)butyl neamine to inhibit protein synthesis, we used the *Escherichia coli* S30 Extract System, Circular DNA kit. In tubes PCR, 0.25 μL of plasmid (pBESTluc DNA) containing the luciferase gene and the regulatory region of the luciferase gene, 2.5 μL of a solution containing all the amino acids, 10 μL of S30 Premix without amino acids (containing rNTPS, tRNA, regeneration of ATP, IPTG, and salts), 7.5 μL of S30 Extract, and 2.75 μL of H_2_O were mixed to obtain a total volume of 23 μL. Then, 2 μL of H_2_O or selected compounds at the MIC against *E. coli* were added (colistin, neomycin B, 3′,6-dinonyl neamine, 3′-heptyl-6-(1-pyrenyl)butyl neamine, all with a MIC of 1 µg/mL). After gently vortexing, the microtubes were centrifuged for 5 s. They were then incubated for 60 min at 37 °C and the tubes were put at 4 °C for 5 min to stop the reaction. The samples were diluted 10 times using the dilution reagent (Luciferase Dilution Reagent). In a 96-well plate, 25 μL of luciferase reagent and 5 μL of each sample were added. The plate was maintained away from light. Luminescence was directly read to spectraMax for 1500 milliseconds with a delay of 2 s before the measurement.

### 2.5. Interaction with LPS—BODIPY-TR-Cadaverine Displacement Assay 

Binding affinity to the cell-free lipopolysaccharides (LPS) from *P. aeruginosa* was determined using BODIPY-TR-cadaverine displacement assay, in which the quenching of fluorescence intensity was observed when the probe bound to LPS and displacement of the probe into the solution led to enhancement of its fluorescence [25,26]. Stock solutions of BODIPY-TR-cadaverine (10 mM) and cell-free LPS (2 mg/mL) were prepared by dissolution in Tris buffer (50 mM, pH 7.4). The assays were performed in 96-well plates. Desired concentrations of BODIPY-TR-cadaverine (final concentration 5 µM) and cell-free LTA (final concentration 5 µg/mL) were mixed and kept for 15 min for complete equilibration. After 15 min, the desired compounds were mixed and the plates were kept for 30 min in the dark at room temperature until equilibration. Further in vitro experiment on BODIPY-TR-cadaverine displacement assay using *P. aeruginosa* ATCC 27853 was also carried out. The assays were performed in 96-well plates. The desired concentrations of BODIPY-TR-cadaverine and freshly grown bacteria (final *A*_620_—0.05) were mixed. The mixture was kept for 30 min in the dark at room temperature until equilibration. After 30 min, the desired concentrations of compounds and mixture of BODIPY-TR-cadaverine (final concentration 5 µM) and freshly grown bacteria were added to the plate and kept for 30 min. The fluorescence intensity was measured on a SpectraMax Gemini XS microplate spectrofluorometer (Molecular Devices Corporation, California, CA, USA) using excitation and emission wavelengths of 580 nm and 620 nm, respectively.

### 2.6. Bacterial Membrane Permeabilization—Propidium Iodide Assay

The bacterial membrane permeabilization was determined with a membrane-impermeable fluorescent dye propidium iodide (PI), which is accessible to permeabilized bacteria only [27]. A stock solution of PI (3 mM in pure water) was diluted 100-fold with the bacterial suspension (*A*_620_ of 0.05). 3′,6-dinonyl neamine along with other compounds of interest, at final concentrations ranging from 1 to 7 µM, was added to the PI-containing (final concentration 3 µM) bacterial suspension in 96-well microplates. The fluorescence intensity was measured with a SpectraMax-M3 microplate reader (Molecular Devices, Sunnyvale, CA, USA) at 25 °C after 15 min of stabilization at excitation and emission wavelengths of 540 and 610 nm, respectively.

### 2.7. Statistics

Statistical processing of the results was carried out using one-way ANOVA with Tuke as a posttest (Graphpad prism, GraphPad Software, San Diego, CA, USA). The results are considered statistically significant when the *p*-value is lower than 0.05 (* *p* < 0.05, ** *p* < 0.01, *** *p* < 0.001).

## 3. Results

### 3.1. 3′,6-Homodialkyl Neamine Derivatives Are Active against Clinical Strains of P. aeruginosa and E. coli

We previously demonstrated the potential of the 3′,6-homodialkyl neamine derivatives carrying branched or unbranched alkyl chains against sensitive and resistant strains of *P. aeruginosa*, *E. coli*, *Acinetobacter lwoffii*, *K. pneumoniae*, and *S. aureus* [21]. We now extended the study to clinical *P. aeruginosa* strains (PA 238, PA 272, and PA 307) and *E. coli* (06AB003; Table 1). The MIC values of the 3′,6-dinonyl neamine and the 3′,6-di(dimethyloctyl) neamine were determined and compared to those previously described for selected resistant *P. aeruginosa* strains and *E. coli* ([21]) or measured for conventional aminoglycosides.

Very interestingly, the homodialkyl neamine derivatives, with or without branched alkyl chains (3′,6-di(dimethyloctyl) neamine and 3′,6-dinonyl neamine), showed low MIC (1 µg/mL) on PA272 and PA307 strains resistant to gentamicin (MIC 32 µg/mL) and tobramycin (MIC 8–16 µg/mL). So, we extended the previous data obtained on *P. aeruginosa* Psa F03 [21]. Similarly, both derivatives showed low values of MICs (2 µg/mL (3′,6-dinonyl neamine) and 1 g/mL (3′,6-di(dimethyloctyl) neamine)) against *E. coli* 06AB003 (MIC > 128 µg/mL (gentamicin and tobramycin)), as was described for *E. coli* PAZ505H8101 (tobramycin) and L8058.1 [21]. Neamine was globally inactive on all selected strains. Against clinical strains, neomycin B showed activity on all the *E. coli* strains selected. On *P. aeruginosa*, neomycin B was active only on PA 406 and PA313. 

### 3.2. 3′,6-Homodialkyl Neamine Derivatives Are Active against β-Lactamases-Resistant Strains

Next, we investigated the antibacterial activity of the 3′,6-dinonyl neamine and its isomer, the 3′,6-di(dimethyloctyl) neamine on selected β-lactamase-resistant Gram-negative bacteria. Table 2 illustrates MICs values for strains expressing β-lactamases, class B (metalloprotéases; i.e., *P. aeruginosa* VIM-2 and *E. coli* NDM-1) as well as class A (i.e., *P. aeruginosa* BEL-1 and PER-1, CTX-M-15 S208/1R2), class C (*P. aeruginosa* AmpC overexpression), and class D (*Klebsiella* OXA-48).

Both homodialkyl neamine derivatives were highly active, with MICs ranging from 1 to 2 µg/mL against the two class B strains, resistant to cefotaxime and meropenem (128->128 µg/mL). In sharp contrast with homodialkyl neamine derivatives, gentamicin and tobramycin showed MICs ≥ 128 µg/mL.

Against class A β-lactamases (currently corresponding to ESBL; [28]), both 3′,6′-dinonyl neamine and 3′,6-di(dimethyloctyl) neamine showed low MICs (1–2 µg/mL) on BEL-1, PER-1, and CTX-M-15 S208/1R2 strains. By contrast, cefotaxime had no activity against *P. aeruginosa* BEL-1 and PER-1 or against ESBL CTX-M-15 S208/1R2, with high MICs (128 µg/mL and 32 µg/mL). Meropenem and gentamicin were still active, with MICs ranging from 0.5 to 4 µg/mL. Neomycin B was inactive against *P. aeruginosa* BEL-1 and PER-1. Tobramycin and neamine were inactive on the selected strains belonging to class A and B β-lactamases.

On class C and D, the two selected homodialkyl derivatives were still active as gentamicin and tobramycin.

### 3.3. MIC against P. aeruginosa of 3′,6-Dinonyl Neamine Derivative Increases Slightly upon Long Exposure in Contrast to the Other Antibacterial 3′,6-Dialkyl Neamine Derivatives

To study, over a long period of time, the effect of incubation of *P. aeruginosa* with homodialkyl neamine derivatives, including those having a branched of alkyl chains on MIC, we determined, over time, the variation in MIC values of 3′,6-dihomoalkyl neamine derivatives bearing chain lengths in C7, C9, and C11 in comparison with ciprofloxacin and of 3′,6-dinonyl neamine in comparison with 3′,6-di(dimethyloctyl) neamine. 

As already demonstrated [21], the MIC of 3′,6-dinonyl neamine against susceptible *P. aeruginosa* ATCC 27853, after exposure to half-MIC at different times over more than two weeks, appeared to be slightly increased from 1 to 4 µg/mL on day 11. In comparison, MIC values measured with ciprofloxacin increased faster from 0.5 to 16 mg/mL on day 7. Here we extended the study to three 3′,6-dihomoalkyl neamine derivatives bearing chain lengths in C7, C9, and C11, for 15 days at antibiotic concentrations that are adjusted according to the evolution of the MIC (Figure 1). 

The MIC of 3′,6-diheptyl neamine and 3′,6-diundecyl neamine on day 15 increased from 16 to 128 µg/mL (8 times more than the initial value) and from 4 to 64 µg/mL (16 times more than the initial value), respectively. In comparison, the 3′,6-dinonyl neamine increased from 1 to 2 µg/mL, whereas that of ciprofloxacin increased from 1 to 16 µg/mL. There is a clear dependency of the hydrocarbon chain length for the increase in MIC over time. 

Interestingly, study for a longer duration (30 days of exposure) confirmed the minor MIC increase when bacteria were incubated with 3′,6-dinonyl neamine and its branched isomer, in contrast with ciprofloxacin (64 times increase in the initial value after 30 days), suggesting no major change in the antibiotic susceptibility during a long-term treatment.

### 3.4. 3′,6-Homodialkyl Neamine Derivatives Are Not Accumulated in Bacteria and Did Not Inhibit Protein Synthesis 

To give an insight into the mechanism of action of homodialkyl neamine derivatives, we investigated their potential ability to inhibit protein synthesis, the primary mechanism of action of aminoglycoside antibiotics, by using an in vitro transcription/translation assay [29] (Figure 2). The effects on bacterial protein synthesis were determined at the MIC against *E. coli*. 

As expected, neomycin B inhibited bacterial protein synthesis. By contrast, 3′,6-dinonyl neamine and its fluorescent analogue 3′-heptyl-6-(1-pyrenyl)butyl neamine [21], which showed similar broad-spectrum antibacterial activities and have close lipophilicity, did not inhibit protein synthesis and showed similar profiles as the ones obtained with colistin, known to act on the membrane [30,31]. These results extended the previous data obtained for dinaphtylalkyl neamine derivatives [29]. The addition of lipophilic groups to the neamine core changes the primary mode of action of aminoglycosides since 3′,6-dinonyl neamine and its fluorescent derivative are unable to inhibit the protein synthesis.

To further investigate the capacity of the antibiotic to pass through biological membranes, we determined the intracellular concentration of the antibiotic potentially crossing the *P. aeruginosa* membranes by measuring the cellular concentration/extracellular concentration (Cc)/Ce) ratio (Figure 3). The experiments were performed at 37 °C and 4 °C to ascertain the potential role of active efflux. At 37 °C, after 5 and 30 min of incubation, the accumulation (Cc/Ce) reached a value of around 2, slightly increasing for longer duration of incubation (60 and 120 min). At 4 °C, the Cc/Ce was stable after the duration of incubation (from 15 to 120 min) and was around 1.5. 

### 3.5. 3′,6-Homodialkyl Neamine Derivatives Interact with Lipopolysaccharides 

The absence of inhibitory effect on protein synthesis in *P. aeruginosa* and the intracellular accumulation of dialkyl neamine derivatives in *P. aeruginosa* confirm the bacterial membranes as the target for their antibacterial activity. Thus, we investigated the potential interaction of 3′,6-dinonyl neamine with lipopolysaccharides [23] (Figure 4). LPS–LPS interactions, together with the bridging of the proximal negatively charged functional groups by metal ions such as Ca^2+^ and Mg^2+^, are critical for the barrier impermeability of bacteria. 

We determined the capacity of homodialkyl neamine derivatives carrying different linear alkyl chain lengths (from C7 to C11) or branched chains to interact with the LPS from the free cell system and from bacteria (*P. aeruginosa*). We used the Bodipy-TR-cadaverine (BC) displacement assay and monitored the enhancement of the fluorescence intensity, reflecting the displacement of Bodipy-TR-cadaverine from its binding to LPS by homodialkyl neamine derivatives, having higher affinity for LPS than Bodipy-TR-cadaverine [25,26]. Polymyxin B (50 µM) was used as a positive control (100%) [30,31,32].

First, the Bodipy-TR-cadaverine displacement in cell-free LPS in the presence of increasing concentrations (0–20 µM) of 3′,6-homodialkyl neamine derivatives was measured (Figure 4A,B). The displacement of the Bodipy-TR-cadaverine from the cell-free LPS increased with an increase in the chain length from 3′,6-diheptyl neamine to 3′,6-dinonyl neamine. No more increase was observed for derivatives with longer alkyl chains. Neamine and neomycin B showed a very weak effect in this respect. At 20 µM, maximum displacement with homodialkyl neamine derivatives (Figure 4C) was ranging from 3′,6-diheptyl neamine (56.8%), 3′,6-dioctyl neamine (76.2%), 3′,6-dinonyl neamine (86.2%), 3′,6-didecyl neamine (90.2%), and 3′,6-diundecyl neamine (90.2%). A comparison between derivatives bearing unbranched alkyl chain (3′,6-dinonyl neamine) and branched alkyl chain (3′,6-di(dimethyloctyl) neamine) (Figure 4B,C) showed maximal displacement values of 86.2% for 3′,6-dinonyl neamine and 92.3% for 3′,6-di(dimethyloctyl) neamine. The maximal effect was reached between 5 and 10 µM.

Second, we extended the assay to the potential displacement of the binding of Bodipy-TR-cadaverine to LPS from *P. aeruginosa* ATCC 27853 by the series of homodialkyl neamine derivatives (Figure 4D,E). At 20 µM concentration of dialkyl neamine derivatives, the maximum displacement (Figure 4F) ranged from 26.9% for 3′,6-diheptyl neamine to 37.3% for 3′,6-dioctyl neamine, 49.8% for 3′,6-dinonyl neamine, 57.9% for 3′,6-didecyl neamine, and 75.5% for 3′,6-diundecyl neamine. The maximum displacement values of 3′,6-dinonyl neamine and 3′,6-di(dimethyloctyl) neamine were 49.8% and 82.3%, respectively. Compared to the cell-free system, the displacement of the binding of Bodipy-TR-cadaverine to LPS was reduced. Moreover, the slope trend was less steep, and maximal effect was observed around 10 µM. Neamine and neomycin B did not displace Bodipy-TR-cadaverine from *P. aeruginosa*-bound LPS. 

To further confirm and characterize the interactions between 3′,6-homodialkyl derivatives, we also investigated the effects of the interactions on LPS micelle size (Figure 5). The average diameter of LPS micelles increased with an increase in the chain length from 7 to 11 in homodialkyl neamine derivatives. Compared to the diameter of control LPS micelles (106 nm), the average diameter of LPS micelles incubated with 10 µM of 3′,6-homodialkyl neamine derivatives increased to 122 nm with 3′,6-diheptyl neamine, 147 nm with 3′,6-dioctyl neamine, 289 nm with 3′,6-dinonyl neamine, 285 nm with 3′,6-didecyl neamine, and 467 nm with 3′,6-diundecyl neamine. The effect on size observed with 3′,6-dinonyl neamine (289 nm) was not significantly different from the effect induced by 3′,6-di(dimethyloctyl) neamine (248 nm). The effect induced by shorter-chain-length 3′,6-diheptyl neamine and 3′,6-dioctyl neamine was weaker than that observed with higher-chain-length dialkyl neamine derivatives (chain length from 9 to 11).

These results show the role of the lipophilicity and the length and the branching of hydrocarbon side chains in the interaction with LPS.

### 3.6. 3′,6-Homodialkyl Neamine Derivatives Induce Membrane Permeabilization 

Lastly, for investigating if the interaction with LPS located at the outer membrane of *P. aeruginosa* could result in inner membrane permeabilization, we monitored the fluorescence of propidium iodide. Propidium iodide is unable to cross intact membranes and fluoresces after binding to DNA. Fluorescence is, therefore, only observed if membranes have been permeabilized. Colistin was used as a positive control [31].

The fluorescence of propidium iodide in the presence of increasing concentrations of 3′,6-homodialkyl derivatives with side chains bearing increasing carbon atoms was illustrated (Figure 6). 

Permeabilization of the bacterial membrane was observed after exposure to all 3′,6-dialkyl neamine derivatives. The maximal effects were very similar irrespective of the compound and around 50–60%. Compared to the other derivatives, the 3′6-didecyl neamine induced an effect at lower concentrations, and even the maximal effect was similar to that observed with the other derivatives. There was no or very low permeabilization with neamine and neomycin B. Regarding the effect of side chain branching, the dose required to obtain 50% of the maximal effect is lower for the branched compounds compared to the unbranched side chains, and even the maximal effect was equal for both compounds. 

## 4. Discussion

For the past few years, in the search for new targets for antibiotics, membranes have been more and more explored [33,34,35]. Huge efforts have been made in the RAS work from the aminoglycoside backbone, leading to appealing lead compounds with antifungal [36,37] or antibacterial activity [38,39,40,41]. Tobramycin [40], nebramine [42], and neamine [43] are the pseudo-oligosaccharide scaffolds most often used for the development of new amphiphilic aminoglycoside antibiotics. Our research focuses on the new amphiphilic derivative of neamine, which is able to bind to bacterial membranes, partition into the membrane–water interface, and alter the packing and organization of lipids and the activity of proteins involved in the cytokinesis and bacterial shape [24,29]. 

Neamines bearing two side linear or branched alkyl chains of various length have been identified as potential antibacterial leads [17,21]. To finely delineate the structure–activity relationships relating the antibacterial activity to a lipophilicity window, we synthesized 3′,6-homodialkyl neamine derivatives with clogP ranging from −13.5 (3′,6-diheptyl neamine) to −10.3 (3′,6-diundecyl neamine) with 3′,6-dioctyl neamine (−12.5), 3′,6-dinonyl neamine (−11.9), and 3′,6-didecyl neamine (−11.1) [21]. A branched analogue of 3′,6-dinonyl neamine, 3′,6-di(dimethyloctyl) neamine, having a close −11.3 clog P, was also synthesized [21].

Here we probed whether the antibacterial activity of these derivatives can be extended to clinical Gram-negative bacteria, ESBL, and carbapenemase-producing strains and the role of length and/or branching of alkyl chain for the emergence of *P. aeruginosa* resistance, as well as the ability to bind to LPS and provoke membrane permeabilization. To acquire more knowledge on the mechanism of action, we also determined the intracellular accumulation and potential effect of protein synthesis induced by a fluorescent derivative analogue from the most promising compound. 

From a pharmacological point of view, we showed similar good activity of 3′,6-dinonyl neamine and its analogue of close lipophilicity, bearing a branch at the end of the alkyl chain, 3′,6-di(dimethyloctyl) neamine, against clinical strains of *P. aeruginosa*, ESBL, and carbapenemase-producing strains. We also showed how the length of the alkyl chain strongly affects the emergence of resistance as evidenced by an increase in MICs after the exposure of susceptible *P. aeruginosa* to half-MIC of dialkyl derivatives for more than 15 days. The lipophilicity value (clogP) of around −11 is mostly suitable for a neamine derivative to show enhanced antimicrobial activity, and it is also very tough for Gram-negative bacteria to generate resistance against them, especially the 3′,6-dinonyl- and the 3′,6-di(dimethyloctyl)-neamine derivatives. By contrast, the steric hindrance does not seem to have an impact on susceptibility to resistance. The large increase in MIC observed with ciprofloxacin could result from multiple resistant determinants against fluoroquinolone antibiotics, such as multidrug efflux systems and pyocin secretion [44,45] encoded by *P. aeruginosa*. The close lipophilicity of the 3′,6-dinonyl and the 3′,6-di(dimethyloctyl) neamine derivatives can probably be related to their binding on and/or and/or insertion into the different membrane components present in multiple copies such as LPS, making the emergence of resistance very difficult.

The mechanism behind these features could be targeting the LPS already described as the main target for polymyxin B nonapeptide [46]. We highlighted the critical role of the length of alkyl chains for the capacity of 3′,6-homodialkyl neamine derivatives to bind to LPS, both to LPS extracted from *P. aeruginosa* or decorating the bacteria. The increase in interaction was in direct relation with the length of alkyl chains and with an increase in clogP. Such fatty acyl chain length specificity could be related to the fact that *P. aeruginosa* lipid A carries 3-OH C10, rather than 3-OH C14 at the 3 position, as for *E. coli*, e.g., [47], even the homodialkyl derivatives are active against *E. coli*. The explanation could be also more complex and the backbone inclination angle of asymmetric penta-acyl lipid A could optimize the position and orientation of the 1- and 4′-phosphates. In addition, changes in chemical composition of lipid A anchor such as acylation and addition of aminoarabinose or phosphoethanolamine moieties on LPS [48] may lead to changes in binding and/or insertion into the outer membrane (OM) and outer leaflet of *P. aeruginosa* (OM).

Moreover, the differences in chain length could contribute to the complementarity between the molecular shapes of the amphiphilic neamine derivatives and LPS. LPS molecules effectively enlarge the reactive surface of the bacteria in relation to the volume. Penta-acyl lipid A domain of the *P. aeruginosa* LPS is characterized by a small tail volume (compared to the hexa-acyl lipid A of *Salmonella minnesota*, e.g., [47]), which could be associated with close packing and peculiar polymorphism in a non-lamellar organization contributing to the efficiency of LPS-dependent protein folding and interaction with drugs [47]. We already reported [23] the critical role of the molecular shape for derivatives grafted with hydrophobic moieties such as naphthylalkyl groups. The change in the spatial structure of the LPS aggregates, particularly the transition from a lamellar into a cubic inverted structure, gives evidence that besides an electrostatically driven adsorption of homodialkyl neamine derivatives on the surface, a subsequent intercalation into the hydrophobic moiety takes place, potentially leading to a change in the molecular shape of the LPS/lipid A molecules [49,50,51]. The molecular shape of each derivative bearing side chains differentiated by length and branching could dictate the way of interacting with LPS and the potential changes induced in curvature and/or activity.

Interaction of amphiphilic neamine derivatives with LPS could result in an effective high concentration of antibiotic within the bacterial outer membrane, leading to potential changes in membrane asymmetry and membrane permeability.

First, an interaction between 3′,6-dialkyl neamine derivatives and LPS might induce the migration of phospholipids from the inner to the outer leaflet of the OM, disrupting the outer membrane asymmetry and leading to changes in LPS synthesis and transport. So, conformational changes in the LptD/LptE proteins located at the outer membranes and implied in the system transport of LPS from the periplasmic side of IM across the periplasm to the cell surface [52] could result from the binding of amphiphilic neamine derivatives to LPS. In turn, this could result in stimulation of the outer membrane vesicle (OMV) formation as described for Pseudomonas quinolone signal (PQS) interacting with lipid A [53,54]. 

Second, an interaction between negatively charged LPS and positively charged antibiotics can modify the barrier penetration, weakening the OM structure [55]. Anionic LPS molecules contribute to the OM’s ability to function as a molecular permeability barrier that protects the bacterium against hazards in the environment [56]. OM constitutes a major hurdle for drug uptake in Gram-negative bacteria, especially for *P. aeruginosa*, which has 12- to 100-fold reduced outer membrane permeability relative to that of *E. coli* [57]. Polyanionic LPS molecules are well ordered because of packing among acyl chains and bridging of phosphate groups by divalent cations, e.g., Ca^2+^ and Mg^2+^ [58,59]. The hydrophilic O-antigen and core sugar groups also contribute to permeability by maintaining a steric shield [59]. A detailed understanding of how the chemical structure of the LPS modulates the macroscopic properties of the outer membrane is paramount for the design and optimization of novel drugs targeting clinically relevant strains [60].

To ascertain if interactions with LPS might modify the barrier penetration to dialkyl derivatives, we characterized their intracellular accumulation resulting from a complex relationship between the active efflux and the permeability barrier [61]. Quantitative techniques able to precisely determine drug accumulation in the subcellular compartments, where their targets are actually located, are still missing [62]. We used 3′-heptyl-6-(1-pyrenyl)butyl neamine. This antibacterial probe was selected for its physicochemical similarity to the 3′,6-dinonyl neamine with similar clogP and absence of an additional group with electron-withdrawing properties as for the dansyl moiety. This derivative has shown similar broad-spectrum antibacterial activity to that of dinonyl derivative [21]. However, whether the substantial bulk added through the pyrenyl moiety affects the interaction with LPS is unknown. Despite the absence of dialkyl neamine derivative accumulation in *P. aeruginosa*, the derivatives induced membrane permeabilization to a small molecule such as propidium iodide. Neither the length nor the presence of branching at the end of the alkyl chain has an effect since they are not dependent upon the chain length of homodialkyl neamine derivatives. Therefore, a relatively similar enhancement of membrane permeability was observed. Only the branched derivative required lower concentrations than those necessary to obtain the same effect with the unbranched compound. Loss of membrane integrity could result from generation of transient imperfections. 

As a perspective, derivatives that bind to LPS with moderate affinity and disrupt the supramolecular organization of the LPS might sensitize pathogenic bacteria to other antibiotics [63,64]. The disruption of the supramolecular organization of LPS could enhance the intracellular accessibility of the antibiotics, thus rendering the micro-organisms susceptible at sub-lethal concentration of drugs. In particular, amphiphilic aminoglycoside derivatives might enhance access to intracellular targets and display synergism with other compounds. Interestingly, homodialkyl neamine derivatives could have, in addition to their own antimicrobial activity, the ability to act as sensitizers against *P. aeruginosa*, as already tested for other compounds in preclinical and clinical studies [64,65,66,67]. If and how changes in LPS conformation and/or structure and Gram-negative peptidoglycan building/recycling are connected is still unknown [68,69]. 

In conclusion, we highlighted the importance of hydrophobic interactions between aminoglycoside derivatives and Gram-negative bacteria membranes, with the interest afforded by the grafting of alkyl chains to the neamine core in 3′- and 6-positions for binding to LPS of the outer membrane of *P. aeruginosa*. Here we demonstrated a relationship between the biological activities of 3′,6-dialkyl neamine derivatives with different linear or branched chain lengths and their lipophilicity. 3′,6-Dialkyl neamine derivatives with a chain length of nine carbon atoms and the branched compound with steric hindrance (3′,6-di(dimethyloctyl) neamine) were shown to be the most effective of the dialkyl neamine compounds. The lipophilicity and the mobility of the hydrophobic moiety of 3′,6-dialkyl neamine derivatives are critical for their effects. Collectively, all results suggested that amphiphilic aminoglycosides derived from neamine are interesting molecules for the development, in the future, of new antibiotics targeting bacterial membranes and/or tools for the study of the dynamics of bacterial membranes. How these activities could be extended to antiviral [70] or antifungal [71,72,73] activities and what are the strategies to block LPS biogenesis [74] should also be explored and could open new avenues based on the current work.

## Figures and Tables

**Figure 1 ijms-22-08707-f001:**
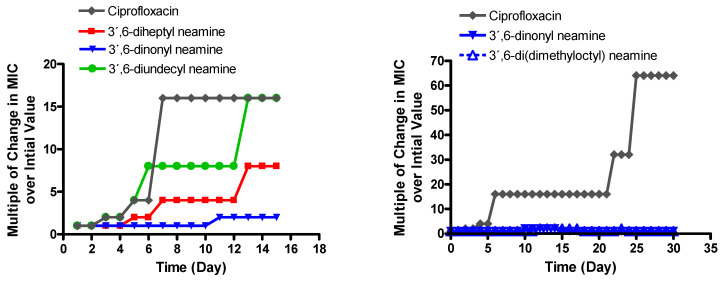
Evolution of the MIC against *P. aeruginosa* of (**left**) 3′,6-diheptyl neamine (C7), 3′,6-dinonyl neamine (C9), 3′,6-diundecyl (C11) neamine, (**right**) 3′,6-dinonyl neamine, and 3′,6-di(dimethyloctyl) neamine, in comparison to the evolution of the MIC of ciprofloxacin after exposure to half-MIC concentrations for the indicated times. The concentration of the antibiotic was readjusted each day to remain equivalent to half the MIC. Results are expressed in changes in MICs over initial value.

**Figure 2 ijms-22-08707-f002:**
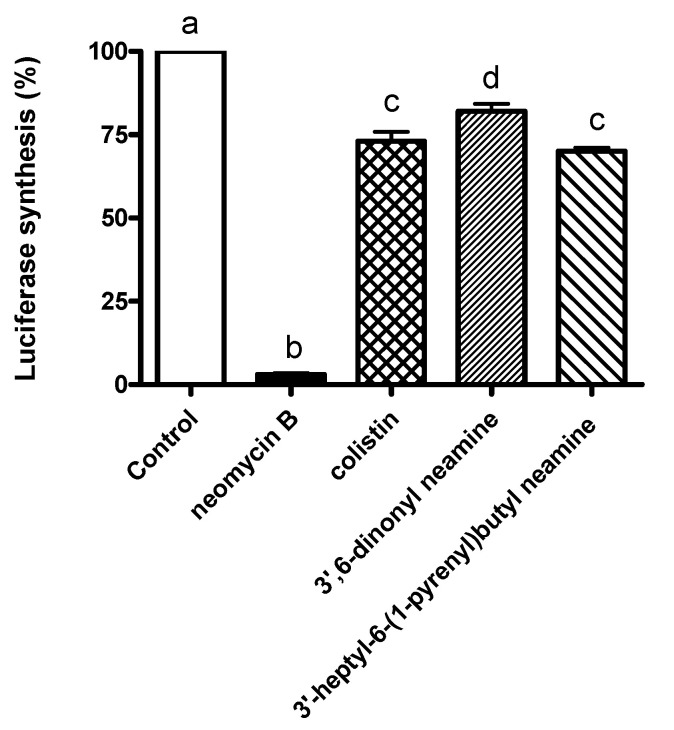
Percentage of luciferase synthesized in the absence (control) or presence (1 µg/mL) of 3′,6-dinonylneamine (0.97 µM), 3′-heptyl-6-(1-pyrenyl)butyl neamine (0.88 µM), neomycin B (1.63 µM) (positive control), and colistin (0.86 µM) (negative control) after 60 min of incubation in a plasmid S30 Extract *E. coli*. N = 2, *n* = 6. Statistical assays were one-way ANOVA analysis with Tukey’s multiple comparison test. In each group, the bars with different letters indicate significant differences (*p* < 0.001).

**Figure 3 ijms-22-08707-f003:**
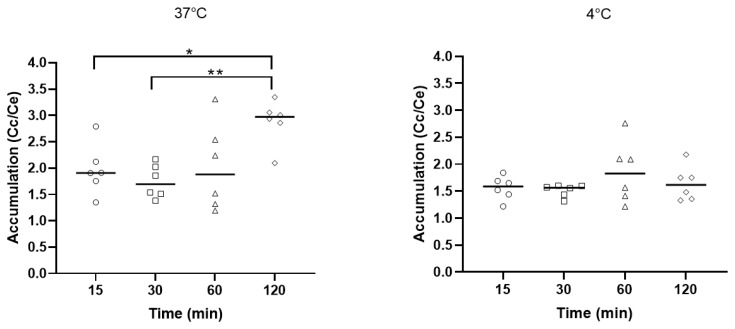
Accumulation factor of 3′-heptyl-6-(1-pyrenyl)butyl neamine derivative at a concentration of 1.7 μM, after 15 (O), 30 (□), 60 (∆), and 120 (◊) min of incubation at 37 °C (**left**) or 4 °C (**right**) in *P. aeruginosa* ATCC 27583. The accumulation factor is obtained by dividing the cell concentration by the extracellular concentration. N = 2. *n* = 6. The statistical difference between the different times is calculated by one-way ANOVA (* *p* < 0.05, ** *p* < 0.01).

**Figure 4 ijms-22-08707-f004:**
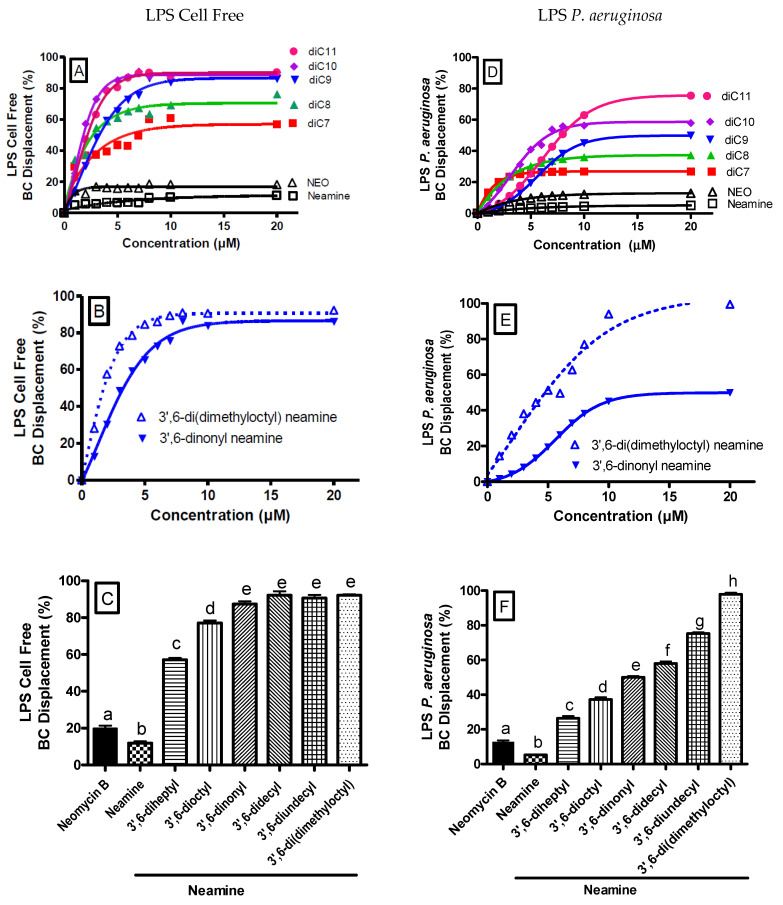
Relative affinity of 3′,6-dihomoalkyl neamine derivatives for cell-free LPS (**A**,**B**) or LPS from *P. aeruginosa* (**D**,**E**) plot for dose-dependent displacement of the fluorescent probe BODIPY-TR-cadaverine from LPS induced by 3′,6-diheptyl neamine (■), 3′,6-dioctyl neamine (▲), 3′,6-dinonyl neamine (▼), 3′,6-didecyl neamine (♦), 3′,6-diundecyl neamine (●), neamine (□), and neomycin B (∆). Fluorescence intensities were normalized as the percentages of probe displacement compared to the probe displacement induced by polymyxin B (50 µM). Error bars are omitted for the sake of clarity (SEM ≈ 4%). (**C**,**F**) Percentage of the maximum effect when the derivatives were at a concentration of 20 μM. Statistical assays were one-way ANOVA analysis with Tukey’s multiple comparison test. In each group, the bars with different letters indicate significant differences (*p* < 0.001).

**Figure 5 ijms-22-08707-f005:**
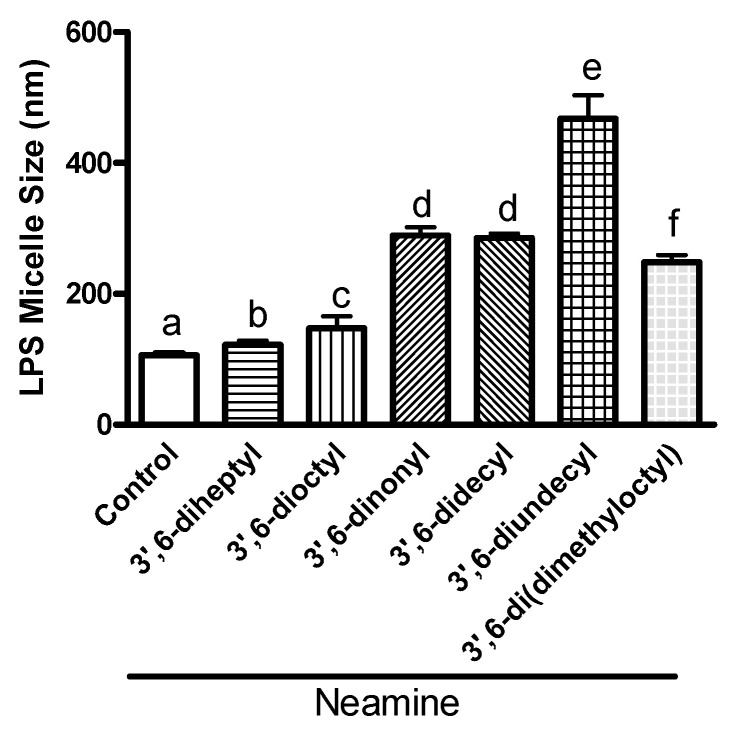
Effect of 3′,6-homodialkyl neamine derivatives on the micelle size distribution of LPS (5 µg/mL) from *P. aeruginosa* in solution (10 mM Tris HCl buffer, pH 7.4). All the compounds were added to a final concentration of 10 µM. Statistical assays were one-way ANOVA analysis with Tukey’s multiple comparison test. In each group, the bars with different letters indicate significant differences (*p* < 0.001).

**Figure 6 ijms-22-08707-f006:**
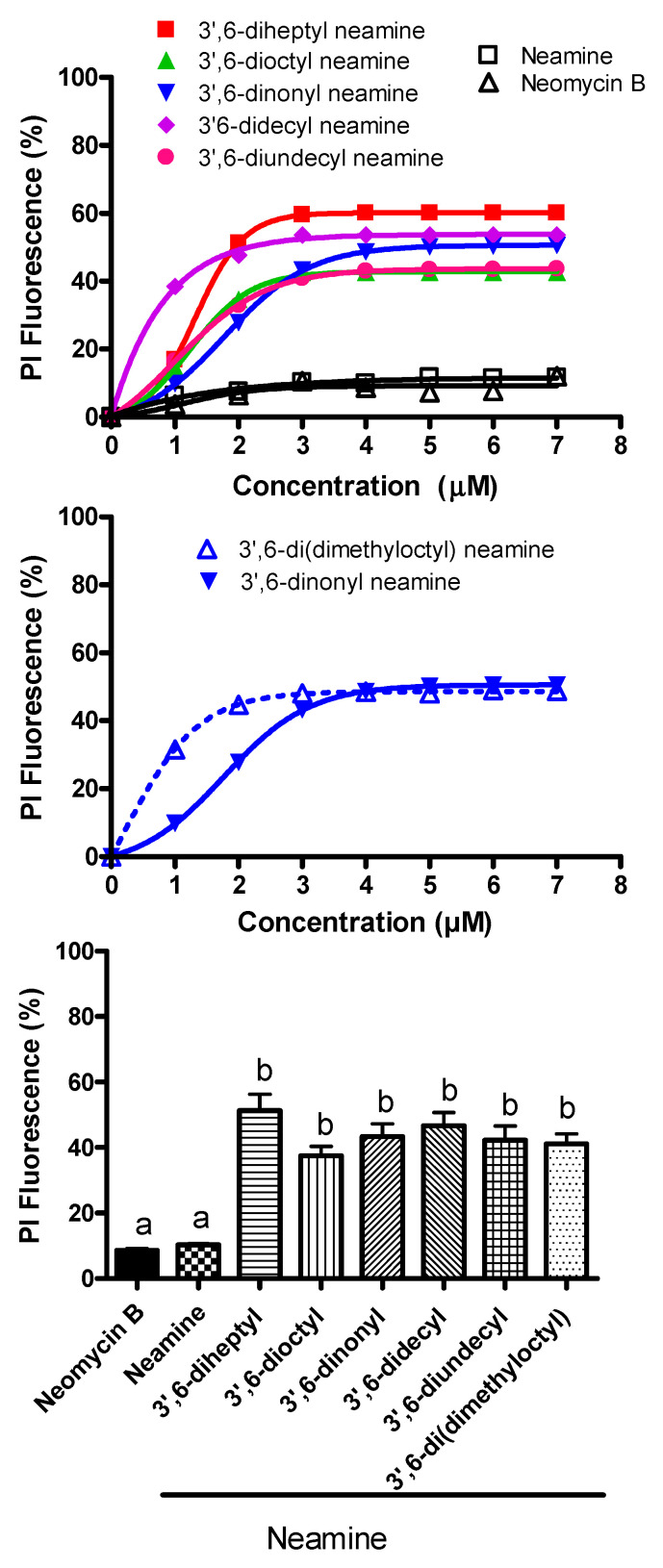
Effect of 3’,6-dialkyl neamine derivatives on *P. aeruginosa* permeability assessed by the increase in the fluorescence of propidium iodide. (Top) 3’,6-diheptyl neamine (■), 3’,6-dioctyl neamine (▲), 3’,6-dinonyl neamine (▼), and 3’,6-didecyl neamine (♦), 3’,6-diundecyl neamine(●), neamine (□), neomycin B (∆), (Middle) 3’,6-dinonyl neamine (▼), and 3’,6-di(dimethyloctyl) neamine (∆). Dose-dependent membrane permeability expressed as a percentage of the maximal value recorded in the presence of 50 µM Colistin, used as a positive control. Error bars are omitted for the sake of clarity (SEM ≈ 4%). (Bottom) propidium fluorescence for 3’, 6-dialkyl Neamine derivatives at 5 µM. Statistical assays were one-way ANOVA’s analysis with Tukey’s multiple comparison test. In each group, bars with different letters indicate significant differences (*p* < 0.001).

**Table 1 ijms-22-08707-t001:** MICs of 3′,6-dinonyl neamine and 3′,6-di (dimethyloctyl) neamine against wild-type and resistant *P. aeruginosa* and *E. coli* clinical strains in comparison with those of gentamicin, tobramycin, neomycin B, and neamine.

Antibiotics	MICs (µg/mL)											
	*P. aeruginosa*								*E. coli*			
	ATCC 27853	Psa.F03 ^a^	PA22 ^b^	PA 406 ^c^	PA 238 ^d^	PA 272 ^d^	PA 307 ^d^	PA 313 ^d^	ATCC 25922	PAZ505H8101 ^e^	L8058.1 ^f^	06AB003 ^g^
gentamicin	<1	>128	4	<0.25	4	32	32	0.5	0.5/1	1	64	>128
tobramycin	<0.25	ND	ND	ND	1	8	16	0.5	1	32	64	>128
neomycin B	32–64	128	32–64	2–4	64	>128	>128	4	1–2	4	1	1
neamine	≥64	>128	>128	64	>128	>128	>128	128	16–32	>128	32	8
3′,6-dinonyl neamine	0.5	4	4	2	1	1	1	1	2–4	2–4	2–4	2
3′,6-di(dimethyloctyl) neamine	0.25	8	4–8	2–4	1	1	1	1	2–4	1–2	1–2	1

^a^ Psa.F03 AAC6′-IIA. ^b^ Surexp MexXY. ^c^ PAO509.5 ΔtriABC, ^d^ clinical strains from Erasmus hospital, Belgium, ^e^ AAC-6′-IB, ^f^ ANT2-IA, ^g^ AAC(3′)-IIa/arm.

**Table 2 ijms-22-08707-t002:** MICs of 3′,6-dinonyl neamine and 3′,6-di(dimethyloctyl) neamine against strains expressing β-lactamases in comparison with those of gentamicin, tobramycin, neomycin B, neamine, cefotaxime, and meropenem. CLSI 2020 breakpoints for *P. aeruginosa*; gentamicin R ≥ 16, tobramycin R ≥ 16, cefotaxime (not defined), meropenem R ≥ 8. *Enterobacteriaceae* (*E. coli* and *K. pneumoniae*); gentamicin R ≥ 16, tobramycin R ≥ 16, cefotaxime (R ≥ 4), meropenem R ≥ 4.

Antibiotics	MIC (µg/mL) ESBL					
	Class B		Class A		Class C	Class D
	*P.aeruginosa* VIM-2	*E. coli* NDM-1	*P. aeruginosa* BEL-1 and PER-1 ESBL	ESBL (CTX-M -15) S208/1R2	*P. aeruginosa* Amp C	*Klebsiella* Oxa-48
gentamicin	>128	>128	4	0.5	<0.25	<0.25
tobramycin	128	>128	64	16	<0.25	0.5
neomycin B	4	2	64	1	2	16
neamine	128	16	>128	16	32	>128
cefotaxime	128	>128	128	32	64	<0.25
meropenem	128	128	4	<0.25	2	4
3,6′dinonyl neamine	1	1	1	2	0.5	4
3,6′di(dimethyloctyl) neamine	1	2	1	1	0.5	2

## Data Availability

Data are available on request to the corresponding author.

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
