# Peer review of "Interest of Homodialkyl Neamine Derivatives against Resistant P. aeruginosa, E. coli, and β-Lactamases-Producing Bacteria—Effect of Alkyl Chain Length on the Interaction with LPS"

_ijms, 2021, doi:10.3390/ijms22168707_

Round 1

Reviewer 1 Report

Line 138- ATCC 27853 strain was...

Line 160 E. coli must be italicized

Line 184- in vitro must be written without "-" and italicized

Line 210 - "the interest " must be replaced with "the potential"

Line 212 -Carefully check that when written for the first time, the names of different species must be given in full

Line 235-"d β-lactamase-resistant"?? maybe 

Line 237- remove the "," from Ambler,....

Line 261- please clarify "To investigate the potential increase of MIC upon duration of P. aeruginosa incubated ???"

Lines 261-285- what is the meaning of these experiments??

Lines 438-349- must be clarified

Line 446- ESBL and carbapenemase producing strains

Author Response

I thank the reviewer for the comments and suggestions for improving the manuscript.

We inserted all of them in the revised version (in yellow in the attached file).

Best regards.

Marie-Paule Mingeot-Leclercq

Reviewer 2 Report

The development of new antibiotics is very important due to the constant "arms race" between humans and microorganisms, however it is becoming increasingly difficult to identify targets in the microorganisms that can be used without having side effects on humans. The present paper describes an interesting approach for new drugs against gram-negative bacteria (the ESKAPE pathogens), which traditionally are more difficult to kill than gram-positives due to the presence of an outer membrane. The manuscript describes assays for measuring the binding of the new drugs to components in the outer membrane, which can be highly useful for systematic screening of other compounds for activity against gram-negative bacteria. The described work is of high quality and adds valuable knowledge to the area.

Author Response

I would like to thank the reviewer for the very positive comment regarding our wok.

Best regards

Marie-Paule Mingeot-Leclercq

Reviewer 3 Report

Authors brought interesting insight into mechanism of action and spectrum of activity of previously described neamine derivatives and confirmed promising properties of drug candidate 3,6 dinonyl neamine. The article is extension and continuation of research discussed in Eur. J. Med. Chem. 2018, 157, 1512-1525. Experiments are well designed and adequately documented. Results are evaluated and discussed correctly. Article is generally well written and after fixing some minor flows I recommend it for publication.

Line 17-18. It is stated that derivatives carrying different alkyl chains C7-C11 were evaluated on different clinical P. Aeruginosa and ESBL strains. This is confusing, because only two compounds were evaluated. C7-C11 derivatives were evaluated only for interaction with LPS and membrane permeability. Please reformulate the abstract.

Author Response

We would like to thank the reviewer for the very positive comments regarding our work.

We modified the abstract as requested. The message is now clearer and more precise.

Best regards.

Marie-Paule Mingeot-Leclercq
